# Review and Assessment of Existing and Future Techniques for Traceability with Particular Focus on Applicability to ABS Plastics

**DOI:** 10.3390/polym16101343

**Published:** 2024-05-09

**Authors:** Ignacy Jakubowicz, Nazdaneh Yarahmadi

**Affiliations:** RISE Research Institutes of Sweden, 40022 Gothenburg, Sweden; ignacy.jakubowicz@ri.se

**Keywords:** ABS plastics, traceability, circular economy, physical marking, recycling

## Abstract

It is generally recognized that the use of physical and digital information-based solutions for tracking plastic materials along a value chain can favour the transition to a circular economy and help to overcome obstacles. In the near future, traceability and information exchange between all actors in the value chain of the plastics industry will be crucial to establishing more effective recycling systems. Recycling plastics is a complex process that is particularly complicated in the case of acrylonitrile butadiene styrene (ABS) plastic because of its versatility and use in many applications. This literature study is part of a larger EU-funded project with the acronym ABSolEU (Paving the way for an ABS recycling revolution in the EU). One of its goals is to propose a suitable traceability system for ABS products through physical marking with a digital connection to a suitable data-management system to facilitate the circular use of ABS. The aim of this paper is therefore to review and assess the current and future techniques for traceability with a particular focus on their use for ABS plastics as a basis for this proposal. The scientific literature and initiatives are discussed within three technological areas, viz., labelling and traceability systems currently in use, digital data sharing systems and physical marking. The first section includes some examples of systems used commonly today. For data sharing, three digital technologies are discussed, viz., Digital Product Passports, blockchain solutions and certification systems, which identify a product through information that is attached to it and store, share and analyse data throughout the product’s life cycle. Finally, several different methods for physical marking are described and evaluated, including different labels on a product’s surface and the addition of a specific material to a polymer matrix that can be identified at any point in time with the use of a special light source or device. The conclusion from this study is that the most promising data management technology for the near future is blockchain technology, which could be shared by all ABS products. Regarding physical marking, producers must evaluate different options for individual products, using the most appropriate and economical technology for each specific product. It is also important to evaluate what information should be attached to a specific product to meet the needs of all actors in the value chain.

## 1. Introduction

During the last decade, EU regulations and the work of the European Telecommunications Standards Institute have encouraged more traceability across the value chain [1], especially in the food and pharma sectors, to guarantee to end users and companies that counterfeiting is being prevented.

Because of the more extensive regulations, more severe security threats to the goods and materials transported along supply chains and the demand for the fast, flexible and secure exchange of information, traceability has become the responsibility of all organizations in the value chain, from raw material providers to retailers who sell products to end customers. The aspects of traceability are unique identification, data capture and recording, link management and data communication. For the implementation of these principles, different technologies have been used, such as automated identification, electronic data processing and electronic data interchange [2].

One of the objectives of the EU-funded project with the acronym ABSolEU (Paving the way for an ABS recycling revolution in the EU) is to propose a suitable traceability system for ABS products through physical marking with a digital connection to a suitable data management system to facilitate automated identification, electronic data processing and electronic data interchange for the circular use of ABS. Using this system is expected to facilitate the selective sorting and recycling of ABS materials as well as the circular use of ABS in general. In addition to this, the system is also expected to provide information by collecting and managing data from different processes throughout the value chain about the origin of feedstock, including the proportion of recycled material, production/manufacturing instructions, service life conditions and disassembly instructions to allow for communication and the interchange of data with actors in the whole value chain, such as consumers, waste managers, recyclers, second-life users and producers, and provide a basis for their decision making. The biggest difficulty in creating a traceability system for ABS materials is the lack of information and data sharing between actors along the value chain and the complexity of ABS plastics.

### 1.1. Characteristics of ABS

ABS resin is a terpolymer consisting of three monomers: acrylonitrile, butadiene and styrene. In general, ABS consists of a continuous phase of copolymers of styrene and acrylonitrile (SAN), whereas butadiene usually forms the dispersed phase. ABS is produced mainly using two different polymerization processes, viz., mass and emulsion processes. In the first of these, styrene and acrylonitrile react in the presence of a polybutadiene substrate, whereas in the second of these, ABS is produced in two steps. In step one, butadiene is produced in an aqueous emulsion using radical initiators and emulsifiers, followed by a grafting step in which styrene and acrylonitrile are emulsion polymerized onto the polybutadiene substrate [3]. The most important mechanical properties of ABS are its impact resistance and toughness. Various modifications of ABS are made to alter its impact resistance, toughness and heat resistance. For example, its impact resistance is heightened by increasing the proportions of polybutadiene in relation to styrene and acrylonitrile, although this causes changes in other properties. In addition to the proportion of polybutadiene, morphology has a significant additional effect on its material properties. For example, increasing the size of rubber particles can increase its toughness, whereas increasing the polymer chain length produces a stronger material [4]. Different manufacturing methods also affect the properties of ABS plastics [5]. In particular, injection moulding conditions should be controlled carefully to ensure that shear is kept to a minimum, to avoid excessive pressures during the packing phase in the mould and to ensure the linear formation of the butadiene chains.

The ageing characteristics of ABS are largely influenced by the polybutadiene content. The thermo-oxidative degradation of ABS leads to the formation of oxidation products and a decrease in unsaturated bonds in the butadiene phase. In addition, cross-linking in the butadiene phase has been observed through studies of molar mass changes during oxidation [6]. Cross-linking and chain scission in the polybutadiene phase are also observed as a result of the photodegradation of ABS but this degradation is mainly limited to the exposed surface of the material [7]. Consequently, it is common to use antioxidants in the composition as well as additives or a surface coating to protect ABS materials against ultraviolet (UV) radiation.

As can be seen clearly from the above description, by changing the proportions of ABS components and additives, ABS materials can be prepared in many different grades for two major manufacturing categories, viz., for extrusion and for injection moulding. Consequently, ABS due to its versatility is used in many applications, such as car parts, toys, electronic housings, consumer products, pipe fittings and many more, and creates the second most amount of waste after polyolefins [8]. The great variety of its uses and the variability in the product composition as well as the presence of polymer mixtures constitute the main obstacles concerning ABS recycling.

### 1.2. Recycling of ABS

Global ABS demand in 2023 was 20.6 million tons, growing at a compound annual growth rate of 4.5% from 2018 to 2023 [9]. This can be compared to the global consumption volume of recycled ABS of 2.6 million tons in 2022 [10]. This means that to realize the European vision of more than half of the plastic waste generated in Europe to be recycled by 2030 (EUR-Lex—52018DC0028), we need to accelerate plastic circularity significantly.

The main problems concerning ABS recycling are the variability in its applications, material compositions, the presence of polymer mixtures and the use of additives. Despite these, there are already European and national requirements on how much material must be recycled within various industrial sectors. For example, one of the biggest areas of use for ABS is the automotive industry. This sector is regulated in the EU by the end-of-life vehicles (ELV directive) [11], which requires 85% of products to be reused and recycled and 95% to be reused and recovered. The new proposal from the EU Commission in July 2023 that aims to enhance the circular design of vehicles requires a minimum of 25% of recycled plastic to be used in vehicle manufacturing, with 25% of that coming from recycled ELVs, which will be achieved soon. The proportion of ABS in plastics that are used in electric and electronic equipment is about 30%. This industry sector is also regulated in the EU by the Waste of Electric and Electronic Equipment (WEEE) directive [12], which requires 70%–80% of waste to be recovered in the form of energy and/or materials.

Mechanical recycling is the most used recycling method for ABS and refers to processing waste into secondary raw materials without significantly changing the materials’ chemical structure. The limitation of the mechanical recycling process is that it may cause chain scission and thermo-oxidative degradation, which degrades the material properties, leading to downcycling [13]. Another common problem is contamination with other polymers, which usually leads to a significant deterioration of the material properties [14]. In addition, even small amounts of some other contaminants can reduce the quality of recycled materials, limiting their end-use options. However, there are also methods to repair recycled materials. The successful renovation of molecular chains and phase interface of recycled ABS was achieved by an in situ chain extension reaction between ABS and pyromellitic dianhydride by modifications via melt blending [15]. Many other additives can be used during the mechanical recycling process to modify/upgrade recycled plastics and make them usable for higher-grade applications.

Another emerging recycling option for ABS is physical recycling, often referred to as solvent extraction. It is a process in which ABS material is subjected to dissolution and purification steps to separate ABS from other polymers and additives, which normally results in pure ABS or SAN. Lu and Chen developed a physical recycling process for toys that produces a recovered ABS suitable for plastic reprocessing if proper additives are added to improve its thermal stability and mechanical performance [16]. Achilias et al. reported that ABS can be recycled from WEEE using the dissolution method. They reported that ABS was dissolved in acetone at room temperature, thus eliminating the risk of thermal degradation [17].

### 1.3. Circular Use of ABS

The circular economy goes beyond the simple concept of material recycling; it approaches a systematic level that integrates economic, environmental and social sustainability [18]. The circular use of ABS materials implies that products and materials are maintained at their highest value for the longest time possible and are designed for the reduction in waste from the outset, avoiding and eliminating hazardous and toxic substances. It also means that manufacturers are willing to introduce recycled materials into their manufacturing processes to limit the number of raw materials used. It is also important to reuse parts from products in the same application as well as convert materials or parts from products at the end of their lives into new applications. This requires the transparent flow of product information and other data between suppliers, manufacturers, logistics, recyclers and customers. Lack of traceability and transparency in value chain transactions have been identified as barriers to the circular use of plastics [19].

The circular approach implies that we need to change the whole life cycle of products and materials, including their design, technological and production systems, distribution, consumption, collection, recycling and final disposal options [20].

### 1.4. Aim and Methods

This article is based on the assumption that traceability has great potential to accelerate the transition to the circular use of plastics. The purpose of this survey is therefore to review and evaluate existing and future traceability technologies that could be suitable for this purpose, in particular, focusing on products made of ABS. Based on this information, we will prepare at a later stage guidelines on how a traceability system for ABS products should be designed to facilitate the transition towards the circular use of ABS products. The innovative value of this paper relies on the fact that it elaborates on the vision of the European Strategy for Plastics in a Circular Economy, which sets ambitious targets for plastic recycling and circularity, within which we are focusing on a very important group of ABS plastics. To our knowledge, it is the first paper that focuses on traceability technologies for products made from ABS.

Our assessment refers to articles, books and works across scholarly and non-scholarly platforms such as Scopus, Web of Science and Google Scholar, European and international standards as well as interviews with stakeholders. We also examined some news articles about the latest developments in the field of traceability. Searches in the Scopus database using a combination of ‘traceability AND ABS’ resulted in 0 publications.

In general, we found a large number of papers that consider different aspects of both materials and information flows as well as technological developments and their evaluation. Our ambition with this article was not a comprehensive literature review of the entire area of traceability. The purpose of this article was to find and evaluate existing and future traceability technologies that could be suitable for use in products made of ABS to enable their circular use. We tried to use the most simplified terms, such as traceability, information sharing, physical marking, plastics and ABS, which resulted in many unrelated results, and we missed several articles from some specific areas of traceability. At the same time, we tried to compensate by using a citation analysis, which gave us additional articles. Due to the limitations of our study, it does not reflect research in the entire field. However, we believe that our article provides valuable information to those who have an ambition to design a product-based information sharing system for a plastic product or product group in general and for ABS products in particular.

## 2. Traceability—A Key to Successful Transition to Circular Economy

### 2.1. Traceability for Circular Use of Plastics

The large variety of plastics used, the varying lifespan of different products and technical challenges in the identification and separation of plastics create difficulties in tracking plastic flows, from the manufacturing of a product to waste management and preparation for the second life of a material. It is generally recognized that traceability has the potential to contribute decisively to improving environmental sustainability and supporting the global objectives of the circular economy. However, the first obstacle many encounter is the definition of this term. There are many definitions of the concept of ‘traceability’ both in international standards and in various dictionaries, and in addition, the respective interpretations of what traceability is are neither precise nor consistent [21]. A reasonably suitable definition can be found in Webster’s Online Dictionary under ‘Environment’ as follows: ‘The ability to trace the history, application, or location of an item, data, or sample using recorded documentation’. This definition can be expanded as follows: ‘Traceability refers to the completeness of the information about every step in a process chain’. In the literature, two different types of traceability are often mentioned, viz., internal traceability and chain traceability. In this paper, traceability refers to chain traceability, which ranges from procuring raw materials through manufacturing, distribution and sale, usage, and handling as waste.

Despite the general belief in the positive effects of traceability especially for the improved recycling of plastics, there is still a lack of reliable and consistent traceability systems. The recycling of plastics is a complex process, which requires a customized infrastructure and cooperation between different actors in a value chain to be able to unleash its full potential. If fully implemented, the benefits for the environment, consumers and society are obvious. The system must contain complex information about the plastic material so that different actors in the value chain can extract the piece of information that is important to them. The information should be attached to the product by labelling it, with possibilities for digital reading, automated identification and electronic data processing. The need for improved material recycling is also mentioned in several publications in which improved sorting processes for mixed plastic flows, increased traceability and knowledge about chemical content are often highlighted as important factors (e.g., The New Plastic Economy, Ellen MacArthur Foundation, 2016).

A crucial role of the traceability system in plastic recycling is ensuring that the new material produced has the appropriate quality for a second life by suitable reusing/recycling processes. For this reason, industry and public institutions implement physical labelling, which verifies all the stages that each batch of product has gone through and establishes its date and place of origin. In addition, the important information that is needed to evaluate the quality of recycled material is the composition, the state of degradation and the level of contaminants, which are not included in any labelling system yet. Moreover, some consumer products made of plastics are subjected in the EU to chemical restrictions, bans, labelling and lab testing requirements. A product label that enables digital information management can also provide advantages during the product’s usage phase. An example of this is a quite recently introduced digital QR marking system (SCANNECT^®^) for tyre manufacturers, in which QR codes are engraved on vehicle tyres for improved internal tyre logistics.

### 2.2. Example of the Scientific Literature in the Field of Traceability

The scientific literature in the field of traceability can be divided into two main disciplines, viz., information sharing (IS) to enable a circular economy and technologies for the physical marking of products able to carry the necessary information, which are particularly important to consumers and recycling organizations. The main research disciplines that contribute to IS are operations research, which studies the economic impact of IS, computer science, which evaluates solutions for reliable and secure IS, engineering, which studies how to record and manage product information throughout a product’s entire life cycle, and environmental management, which assesses the environmental impact of IS [22]. A good example of operations research is a recently published article reporting on a tool that has been developed for economic analyses and financial risk assessments to assist actors in inter-organizational circular systems in evaluating benefits throughout the different phases [23]. Hsieh et al. refined the existing framework that is applied by circular economy supply chains by screening out key factors that are applied to remanufacturing products. They concluded that “optimizing the production process”, “effectively tracking and recycling products”, “redesigning remanufactured rubber products” and “improving resource efficiency” are the criteria that need to be considered first in the manufacturing of remanufactured products [24]. The impacts of materials on human health and the environment are of great significance. The complex interactions between materials and the environment require special consideration in the design of databases. Amos et al. have presented two data processing methods that can be integrated into the experimental process and can be applied across multiple subfields of material science [25]. Bindel et al. have described the development of a system capable of supporting the collection and visualisation of product data that are used in life cycle monitoring systems for electronic products. They also determined what opportunities and barriers exist for embedded information in this domain [26]. However, the general conclusion from our review of the scientific literature on IS is that it is fragmented, and research and practise are disconnected. Thus, to fully explore the value of IS to enable a circular economy, the scientific literature in this multidisciplinary field must be synthesised, and a common understanding must be established [27].

## 3. Examples of Labelling and Traceability Systems Currently in Use

### 3.1. Society of the Plastics Industry (SPI) Codes

In 1988, the SPI developed the international resin identification coding system, which is a set of symbols placed on plastics to identify the polymer type. The primary purpose of the codes was to allow for the efficient separation of different polymer types for recycling as well as to help consumers better understand some differences between materials and products as well as how they affect one’s health and the environment. The recycling symbols are designed as a triangle formed by three circling arrows. The number in the triangle indicates the type of plastic. SPI codes 1 to 6 denote specific commodity resins, whereas number 7 is used to designate miscellaneous types of plastic, including ABS, that are not defined by the other six codes (see Figure 1). The labelling system is described in the newly revised ASTM international standard ASTM D7611 [28].

A major disadvantage of the system concerns the fact that the symbols are not trademarked and therefore not subjected to official regulation for their use. As a result, they can be applied to any item and do not necessarily indicate recyclability. Another disadvantage is the lack of specific symbols for engineering plastics and new materials such as copolymers (e.g., ABS), as well as bio-based, biodegradable and compostable plastics. More recently, the logo of ABS, or the number 9 surrounded by arrows, has been used to denote ABS plastic, as shown in Figure 1.

### 3.2. EU Medical Device Regulations (MDRs)

One of the mandatory traceability systems is applied on medical device products, for which, according to EU Regulation 2017/745 (EU MDR) [29] medical devices must demonstrate that they meet legal requirements to ensure they are safe and are used as intended. It is mandatory to affix the medical device symbol, which shows that the product is a medical device. All the labelling requirements are detailed in MDRs, including the symbols to be used in the labelling of medical devices and their packaging, the trade name and the device’s original name. All the symbols covering the required information in the labelling of the device and the documents such as booklets and manuals that accompany it must be included. The main purpose of labelling is to provide safety information to users such as healthcare professionals, consumers or other relevant persons. If a device carries a CE mark which confirms that it meets the basic requirements, it must be allowed in all EU countries.

Most medical devices must also be marked with a unique device identifier (UDI), which is a unique code that identifies and tracks medical devices throughout their distribution and use. The UDI system was established by the US Food and Drug Administration to improve patient safety and provide a consistent way to identify medical devices. A significant role of a UDI is to simplify traceability, facilitate corrective measures on the market and make product counterfeiting more difficult. The use of the UDI system can also improve purchasing and waste disposal policies and stock management by health institutions, recyclers and other operators, as well as ensure compatibility with other authentication systems. Figure 2 below illustrates a typical label on a medical device.

### 3.3. EU Toy Safety Directive (TSD) (Directive 2009/48/EC)

A traceability system is also applied on toys for the EU, where all toys must comply with the TSD and must carry a CE mark. This mark is an indication that the toy meets the safety requirements. The directive sets out the labelling requirements for general toy products, including plastic toys. In addition to the CE mark, the required labelling information includes product traceability information (e.g., the manufacturer’s location and contact information, batch ID, model number and date of production) as well as all relevant details related to the sourcing of the product, or the raw material used in making the product. Figure 3 below illustrates a typical label on a toy.

### 3.4. EU Electronic Product Regulations

A traceability system for electronic and electrical products is mandatory in the EU, and products intended for the EU market must meet requirements in regards to safety standards, labelling, documentation and testing. Products covered by one or more EU directives (LVD, RED, EMC, RoHS, etc.) must carry the CE mark, which indicates conformity with the requirements. The CE mark must be permanently affixed to the product and its packaging. Importers also need to attach a permanent traceability label to the products and packaging, showing information about the company and the product. Both importers and manufacturers also need to provide documents such as the Declaration of Conformity, user instructions and technical files showing that the product is compliant with certain product directives and standards. Figure 4 below shows an example of electronic product traceability labelling. The information is limited, non-specific, and effective for waste management or the recycling sector, though it could be supplemented with appropriate information to help some actors in the electronic supply chain.

## 4. Information-Based Traceability of Plastics

Today’s traceability systems for plastic products are not designed to provide appropriate information to all actors in a value chain; the reasons for this are many and depend on the product. Some of the reasons are related to a lack of collaboration between partners in a value chain, the poor utilization of new technical possibilities and an insufficient understanding of the benefits of traceability. The most challenging parts are understanding and acceptance of the benefits of traceability for the environment, society, a circular economy and individual actors. To achieve the set goals for plastic recycling, a suitable traceability system must be introduced with broad acceptance from all players along the value chain and that is clearly supported by society. In the near future, traceability and information exchange between all actors of a value chain in the plastics industry will be crucial to establishing more effective recycling systems. A suitable traceability system should combine physical marking with data sharing platforms that provide access to information about products and materials. A recent scientific review paper analyses four different technological areas for information-based tracking solutions for plastic materials, viz., physical markers and tracers, blockchain, Digital Product Passports and a certification system [30]. The authors summarized the strengths, weaknesses, opportunities and challenges for all four tracking technologies in an SWOT analysis, as summarized in Table 1.

However, in our opinion, data sharing systems and physical marking should be discussed separately. For data sharing, there are various digital technologies that enable information that is attached to a product to identify it as well as store, share and analyse data throughout its life cycle. This is described more in Section 4.1. A physical mark must be applied directly onto the plastic material, using a label on the product’s surface or by the addition of a specific material to a polymer matrix which can be identified at any point in time with the use of a specific light source or device. Physical marks can be accomplished with several different methods as described in Section 4.2.

### 4.1. Data Sharing Systems

In a circular economy, actors must be able to access data, information and knowledge from across the entire value chain, meaning they must have access to an advanced data sharing system that can allow for the exchange of relevant data without trespassing on another actor’s intellectual property rights. Currently, there are still barriers because of historically weak networks and communication. Data sharing systems are digital platforms for information exchange such as Digital Product Passports, blockchain or standard and certification systems. A well-functioning system makes it possible to make reliable information about a product available throughout the value chain. The shared information about regulatory compliance, value chain efficiency, material composition, environments of use and consumer engagement is crucial to understanding the full life cycle of a product and to facilitate separation, recycling and valorisation techniques.

#### 4.1.1. Digital Product Passport

A Digital Product Passport (DPP) is a tool for collecting and sharing product data throughout its entire life cycle. It is used to provide information about a product’s sustainability as well as its environmental impact and recyclability. A DPP creates ‘a digital twin of a physical product and securely records event, transactional and sustainability-based data from across the product’s life cycle. The digital twin is commonly associated with the physical product via a QR code, barcode, or other technology—with the DPP being accessible via a smart device application or similar’ [31]. The expected benefits of a DPP are access to data, flexibility, transparency and the accountability of customers.

However, many challenges remain such as different interests of stakeholders, missing underlying standards, missing infrastructure, data ownership and intellectual property rights. Other challenges include balancing transparency with the safeguarding of sensitive information and the use of centralized databases that store copies of data, which can easily become outdated, leading to poor data quality.

#### 4.1.2. Blockchain

By definition, a blockchain is “a distributed database that maintains a continuously growing list of ordered records, called blocks”. These blocks are linked using cryptography. Each block contains a cryptographic hash (a digital fingerprint or unique identifier) of the previous block, a timestamp and transaction data. A blockchain is “a decentralized, distributed, and public digital ledger that is used to record transactions across many computers so that the record cannot be altered retroactively without the alteration of all subsequent blocks and the consensus of the network” [32].

Blockchain solutions enable businesses, regulatory bodies, governments and consumers to exchange information and cooperate to improve and facilitate waste management. It offers the highest level of transparency, data integrity and security, and for these reasons, it is regarded as one of the key emerging technologies for Europe by the European Commission. The commission has therefore released a blockchain strategy to describe Europe’s digital future [33]. There are several already existing initiatives in the area of plastics that utilize blockchain solutions. One of these is reciChain, a solution created by BASF in Sao Paolo, Brazil, in which circularity was successfully demonstrated in 2020 by tracking a product’s life cycle “from pellet to pellet”. ReciChain comprises two technology components, viz., a physical tracer that enables the connection of plastic to a digital twin and a blockchain marketplace which creates and translates the digital twin, providing a secure, auditable transfer of ownership and assigning incentives [34]. Gopalakrishnan et al. [35] propose a blockchain-based solid waste management model that can help to enhance the efficiency of waste management. They also estimated the cost aspects associated with blockchain implementation based on several use cases obtained from companies providing blockchain solutions. Arora et al. presented a framework that focusses on the use of blockchain technology from the entry to the exit of any process in a system as well as the identification of important areas to which blockchain technology can be applied and implemented [36]. El-Rayes et al. performed a thorough evaluation of the status of plastic management in research. They found a shortage of publications when the term “blockchain” was combined with six selected areas, viz., “supply chain”, “sustainability”, “environment”, “finance”, “fintech” and “plastic”. Furthermore, among the few documents found in the area of “plastic and blockchain”, only 3% mentioned the circular economy [33,37]. 

An important prerequisite for a successful implementation of blockchain technology is that it must mirror physical reality to be credible. This means that in production, each batch of materials needs to be audited by an independent third party that can verify that the material’s properties, its origin, composition, process and sustainability claims are consistent with its real conditions.

#### 4.1.3. Standards and Certification Systems

Certification systems ensure compliance with standards, rules and regulations. The most reliable certification systems and standards for a circular economy rely on annual auditing to ensure that a manufacturing company has quality procedures that guarantee that its products meet the set requirements. An older example of a standard for recycling is The International Resin Identification Coding System described in Section 3.1. A new European certification system for recycled post-consumer plastic waste is the EuCertPlast Certification. The certification system is based on EN 15343:2007 [38], which specifies the procedures needed for the traceability of recycled plastics [38]. As a complement to EuCertPlast, there is the Recycled Plastics Traceability Certification, which is a certification scheme that provides both evidence of the traceability of recycled plastics from the source and the specific recycled content in products [39].

There is an ISO standard for the computer-interpretable representation and exchange of product manufacturing information with the potential to be used for the purpose of promoting materials recycling, but so far it has not been used for this sole purpose. The ISO 10303 [40] standard ‘Industrial automation systems and integration’ is a comprehensive and modular standard that defines product data, such as geometry, topology, materials and properties, and also specifies the methods and protocols for exchanging product data between different systems, such as CAD and CAM. However, there are major obstacles to this standard being able to be generally accepted and implemented by the plastics industry’s value chains. In particular, SMEs, which are often part of the value chain, and consumers will likely not be able to use the standard due to its complexity and diversity, which makes it difficult to understand, implement and maintain. In addition, it is not always compatible or compliant with existing software applications or platforms used by the plastics manufacturing and recycling industry.

In current standards and certification systems, the traceability of individual products is only possible in rare cases, and the information provided by the system does not cover all aspects needed to facilitate recycling. To increase the usage of standards and certification for traceability, it is important to increase their broad acceptance along the value chain and to speed up the implementation process.

### 4.2. Methods for Physical Marking of Plastic Products

The physical marking of plastic products is an important part of the design for the environment. Plastics must be easily identified and then separated at the end of their life according to their material type, chemical structure and quality to maximize their value in their next usage. Permanent marking is the most desired option for all actors who want to track information about a product. Permanent marking or a physical tracer can be accomplished with several different methods, as described below. However, it is necessary to install relevant equipment within sorting facilities and to establish databases to retrieve information from physical markers for all marking technologies. Moreover, it is important that the readability of the tracer during a product’s entire service life is robust even when the material becomes waste.

#### 4.2.1. Photoluminescent Labelling

Photoluminescence is a well-known phenomenon that involves the emission of light from a material after the absorption of electromagnetic radiation. The process is initiated by photoexcitation with subsequent relaxation processes in which other photons are emitted. Because specific chemical substances emit light of unique wavelengths, this technique can be used as a robust method for the identification and separation of waste plastics, without interfering with other sorting techniques. There are three main types of labels depending on the excitation source, viz., UV−vis, NIR and X-ray radiation.

UV–vis markers can be inorganic chemical compounds such as metal oxides or organic compounds containing aromatic components. In some cases, already-existing NIR light sources can be used as the excitation source. This is because some chemical compounds can undergo a so-called photoluminescent up-conversion. This is accomplished by a two-photon absorption process through an intermediate excited state, leading to the emission of radiation of a higher energy level than that absorbed [41]. The most studied NIR markers are inorganic crystals. However, black-coloured plastics present a major issue and remain an obstacle for waste sorting using this technique. The third type of label works through X-ray fluorescence which can be used to identify specific elements in a sample. Because commodity plastics are hydrocarbon based, it is necessary to insert a chemical compound into a plastic with unique identifiable elements for the method to work. In addition to emission wavelengths, photoluminescence can also be used to identify a marker by measuring the emission lifetime. The emission lifetime is determined by the exponential decay of the intensity of the emission when an excitation source is removed from a material.

A particularly interesting photoluminescent-based sorting system uses carbon quantum dots (CQDs) in a plastic matrix through the formation of a polymer nanocomposite. CQDs are carbon nanoparticles which are less than 10 nm in size and have some form of surface passivation. These nanocomposites can respond to stimuli by emitting different fluorescence wavelengths in the range from the UV to the NIR region and thus can monitor many material parameters due to their wide sensing range. However, more studies are needed to show if these nanomaterials can be incorporated into polymers using scalable manufacturing methods, such as extrusion, instead of the currently used solvent-based methods. A recent review summarizes progress regarding the ‘green’ synthesis of CQDs [42].

Labelling plastics using photoluminescence has great potential to facilitate waste sorting and to improve both the quantity and quality of recycled plastic materials. However, it is quite a new approach and still needs to be standardized and implemented across the whole plastic value chain with the active participation of all its actors. Initially, this technique could be used in the recycling of especially valuable plastics or for products which are used in high volumes, such as in packaging.

#### 4.2.2. Digital Watermarks

Digital watermarks are invisible optical codes embedded in a material that can carry a wide range of product information and can be identified in a sorting plant via optical sensors.

Between 2016 and 2019, various stakeholders from the packaging value chain gathered in a project named HolyGrail to investigate possibilities for improving recycling using chemical tracers and digital watermarks. The conclusion of the project drawn by the large majority of stakeholders was that digital watermarks are the most promising technology. In the follow-up project, HolyGrail 2.0, a semi-industrial testing was performed that mimicked real-life conditions. Characteristics such as detection efficiency, ejection efficiency, purity, prototype stability and routine function, ease of programming the sorting operation and counting capabilities of the prototype were evaluated [43]. The project was a success and led to other potential opportunities, such as the creation of smart packaging and its use in other areas. These include enhanced consumer engagement, linking to a data sharing platform, supply chain visibility, retail operations and more.

#### 4.2.3. Laser Marking Systems

A laser is a device that emits coherent light that allows a beam to be focused to a tight spot. The beam is produced through a process of optical amplification based on the stimulated emission of electromagnetic radiation. Laser marking is a broad category of methods to produce marks on an object using beam exposure. A laser creates various surface changes which can include colour changes due to chemical/molecular alterations, charring, foaming, melting, ablation, photoreduction and more. A variety of laser devices and media are used. The differences depend on how the laser is generated and what beams are produced. A critical factor is of course the legibility of the mark which can be affected by the mark contrast width of the surrounding area. Because plastics have a wide range of colours, surface structures, densities and material designs, specific lasers work better with different materials. It is thus important to choose the right laser (in regards to energy density, pulse width and wavelength) relative to the physical and chemical properties of the material, e.g., the absorptivity. For example, a green laser (532 nm) has a very high absorption ratio that can produce suitable marks on PVC, ABS and PS. An ultraviolet laser (375 nm) is a good replacement for ink marks on HDPE material. Fibre and vanadate lasers (1064 nm) work well on many high-density plastics, such as POM, ABS and many engineering plastics. Infrared and visible lasers can induce black and white changes by the carbonization or foaming of samples, respectively, and other colour changes can be laser-induced on pigmented polymers [44].

The laser marking of plastic products is durable and flexible and does not have any negative effects on the material or environment. The marking process can be applied to a wide range of materials by selecting an appropriate laser beam. Furthermore, the numerically controlled beam motion can create a wide variety of marks directly using a computer. Most of the plastic materials can be laser marked, but if a plastic does not have that ability, it can usually be modified with a suitable additive. Thus, the results of laser marking on plastics depend not only on the type of plastic but also on the specific additive used [45]. Suitable laser marking additives create better opportunities to produce high-contrast marks. The lasers that are most commonly used for marking plastics are the following:

A **CO_2_ laser** produces a beam of infrared light with wavelength bands at 9.6 and 10.6 μm. It is suitable for marking most plastics, such as PE, POM, PC, PP, PET and ABS. The beam only removes a surface layer, leaving a permanent mark. CO_2_ lasers have a very low beam divergence (the angle at which a laser beam spreads out from its source), which is a big advantage if a very fine, detailed mark is needed [46].

An **excimer laser (exciplex laser)** is a form of a UV laser (at 193 or 248 nm) that makes patterns on or etches a plastic product. The marking speed depends on the mark complexity and the depth of the mark, which is created as a series of dots. Excimer lasers are commonly used in the production of electronic devices. The UV energy from excimer lasers causes high levels of electron excitation in irradiated materials, leading to bond breaking and photoablation. Excimer lasers have high-energy photons that can induce photochemical reactions, which may result in colour changes with negligible thermal side effects [47].

**Nd:YAG** (**neodymium-doped yttrium aluminium garnet**; **Nd:Y_3_Al_5_O_12_**) is a crystal that is used as a lasing medium for solid-state lasers. Nd:YAG lasers typically emit light with a wavelength of 1064 nm in the infrared region and are one of the most common types of laser, being used for many different applications, e.g., in manufacturing for engraving, etching or marking a variety of metals and plastics. Nd:YAG lasers are also used to make subsurface markings on transparent materials, such as glass or acrylic glass, and on white and transparent polycarbonate for identity documents. Clemente et al. used a frequency-tripled Nd:YAG laser emitting at 355 nm to produce aesthetical marks on white ABS. They found that the marks had a good level of scratch resistance similar to that of pad-painted marks and a good level of resistance to chemical agents and climate tests [48].

A **fibre laser** is a laser in which the active gain medium is an optical fibre doped with rare-earth elements such as erbium, ytterbium and neodymium. They are related to doped fibre amplifiers, which provide light amplification without lasing [49].

#### 4.2.4. Printing

There are several specialized techniques for printing on plastics, such as inkjet, screen, flexo, UV litho, pad printing and hot stamping, and each one of these requires expertise to ensure the best results. One of the most important features of printing is the tendency of ink to remain attached to a substrate when acted on by different forces such as mechanical (rubbing or abrading) and environmental (sunlight, heat, moisture and chemicals). Adhesion is the result of the physical and chemical interaction between the ink and the substrate in which the contact surface area between the ink and the substrate is the major factor in adhesion. Chemical interactions are influenced by the substrate and ink’s chemical compositions which are specially designed for different plastic types. Two of the most used printing methods for plastics include inkjet printing and pad printing.

##### Digital Inkjet Printing

Inkjet printing can be used to print traceability information, variable data, barcodes, multiline codes and more. It is a modern printing method that uses digital files containing information to be printed, which are created using computer software. An inkjet printer managed through digital files ejects droplets of ink onto the plastic surface to create a mark, image or text. An inkjet printer requires the nozzles to size drops precisely with a high degree of accuracy. High-quality inkjet printing systems must simultaneously integrate printheads, electronic controllers, a suitable pretreatment of the surface and curing. All items must work together to produce the intended results. Used correctly, inkjet printing is fast, flexible and can produce high-quality marks. However, there are some concerns that must be considered, such as the following:operating costs over time because of the expendable items involved;harmful chemicals that can add environmental challenges;ink is not permanent even if it can be made very durable;some chemicals can break down even durable inks, and the mark can be lost.

##### Pad Printing

Pad printing, or tampography, is a printing process that allows for the transfer of complex, detailed graphics onto flat or irregularly shaped surfaces. The process uses an image that has been created on a printing plate which then is coated with a layer of suitable printing ink. After the excess ink has been removed from the plate, a thin film of ink remains in the image. Then, a silicone pad is pressed onto the plate to pick up the ink. The pad is then pressed against the surface of a plastic product, transferring the ink. The process ends with the drying procedure until a full cure. Pad printing is often chosen because of its versatility, accuracy and cost-effectiveness and because it can be used on a variety of plastic materials, including ABS. However, there are some factors that must be considered. The first one is the necessity of removing any impurities from the surface that might affect ink adhesion. The second is the need to use a pretreatment of the plastic surface, such as primers, flame, corona or plasma. The third is the need to use, in some cases, a heat source such as an oven to speed up the ink drying process and chemical reaction [50].

#### 4.2.5. Scribe Marking

Scribe (scratch) marking is a mechanical process whereby a mark is produced using a scribing pin with a carbide or diamond tip that penetrates and indents the surface and creates a continuous and homogeneous line. The amount of pressure the pin applies determines the depth of the mark in the material. Even very lightly scratched characters can be achieved with this method. A scribe marker can work on most plastics, including ABS, among many other types of plastic materials. Scribe marking machines are computer-controlled and are specifically designed to be used on a variety of part shapes and sizes. They work satisfactorily on flat surfaces as well as on round, curved, convex and concave parts. Scribe marking systems are not portable and are quite expensive; however, they are less expensive than laser marking systems but are significantly slower [51].

#### 4.2.6. Dot Peen Marking

Dot peen marking uses rapid vibrations to produce marks. Pneumatically or electromagnetically powered carbide or a diamond stylus assembly is stroked against a surface creating depressions or indentations, which results in a succession of dots with negligible separation.

The markings are precise and thus complex shapes and data matrices can be produced on plastics. The equipment is inexpensive, and the process requires few consumables and little maintenance. Its disadvantages include the amount of noise being made by dot peen markers because of the vibrations, and the resolution of the markers needs to be considered if very clear markings are required [51].

#### 4.2.7. In-Mould Marking

Date stamps have been used for many years in the injection moulding process to enable the traceability of plastic items. A date stamp is a cylinder-shaped metal which is composed of an outer ring and an inner arrow. The stamp looks like a clock with an outer ring containing letters, numbers and other characters. The inner arrow can turn to indicate different positions on the outer ring. In this way, a permanent mark is created, showing information such as the manufacture date, lot numbers, types of plastic material and other information depending on one’s needs. Manual date stamps are operated by operators responsible for manually changing the date stamps. It is sometimes difficult to see to which position the stamp needs to be rotated. If they miss the mark and the stamp moulds show incorrect data, a part may be deemed unusable. These problems can nowadays be avoided by automating the process and using a reliable dating system, which allows the process to continue operating while providing accurate, real-time information. This is achieved through the use of a small electronic control unit that is mounted on the interface to enable the programming of the daters as well as to monitor cycles. In-mould marking also includes the permanent marking of a QR code on a plastic product during injection by inserting a QR stamp into the mould. This is an easy and cheap way to mark plastic products directly. The QR code can contain information about the installation manual, maintenance method, material composition and handling of waste.

#### 4.2.8. Stickers

Stickers are basically pre-printed labels that are fixed by a pressure roller to the surface of a product. However, choosing the right label can be a big challenge because the label materials must be precisely matched to the respective application. First, it is important to take into consideration the surface properties of a plastic product as well as the areas of application. For instance, the surface tension of the plastic plays an important role. Consequently, labels with a rubber type of adhesive adhere better to low-surface-tension plastics than many acrylic adhesives, whereas labels with an acrylic type of adhesive are more suitable for high-surface-tension plastics. Adhesion requirements vary greatly depending on the product and customer requirements and range from very durable with an ultra-aggressive adhesive to removable stickers. The latter are made using pressure-sensitive adhesives which form a bond with the medium without a chemical reaction to facilitate its removal and without leaving any sticky residue behind. Of course, the subsequent areas of application must also be considered. For example, if the labels will be exposed to UV light or if they must be chemical-resistant or seawater-proof, it is important that the legibility of the labels remains satisfactory throughout the entire lifetime of the product.

## 5. Concluding Remarks

A life cycle approach to the use of plastics allows people to make informed decisions that can protect human health and reduce environmental impacts. This approach has the potential to reduce resource consumption, improve the performance of products and extend their lifespan in each life cycle stage. There is a general consensus that correctly designed information-based traceability systems can support the circular economy of plastics, remove obstacles, improve their quality and increase the number of recycled plastics. However, the detailed examination of the current situation regarding plastic waste destinations recently performed by Hsu et al. shows that plastics in the EU are still far from being circular [52]. In the scientific literature, there are many new proposals and developments in information-based technologies and technologies for the physical marking of plastics. Our conclusion is that it is possible to find a suitable combination of the smartest physical and digital tracking technologies to establish material tracking system sustainably for various applications in the market. All ABS products could use the same data management system, most probably the blockchain strategy, which has been pointed out by the European Commission as Europe’s digital future [33]. The situation is different when it comes to the choice of physical marking. Which technology is the best option depends on the product’s size, appearance, manufacturing method, area of use, price sensitivity and more. The situation is also complicated when considering what information a product should carry. Some products need to be tracked even at the individual level to enable the reuse of some parts (e.g., car parts and electronics), whereas other products need specific information to facilitate recycling.

Specially designed information to choose the right recycling method and a possible need for upgrading are also important because the recycling of ABS materials presents big technological challenges due to the variety of material compositions, applications, periods of use and usage environments. In our opinion, the most effective way to achieve the circularity of ABS products is by providing necessary, reliable and easily accessible information to all stakeholders in the value chain of an ABS product. To provide guidelines on how a suitable traceability system for ABS products containing all necessary information should be designed, it is important to have state-of-the-art knowledge of current techniques. This paper has therefore reviewed and evaluated various existing, emerging and future techniques for traceability with a special focus on ABS plastics. Our goal is to identify the most suitable techniques for the identification of ABS materials by marks connected to information-based platforms that enable data management and access to information for the actors along the value chain. Additionally, these techniques enable selective sorting, consumer communications, processing instructions, information about the origin of feedstock and more. Ultimately, the decisive step in the introduction of a traceability system will be the acceptance and involvement of all actors in the ABS plastic industry and society in general.

## Figures and Tables

**Figure 1 polymers-16-01343-f001:**
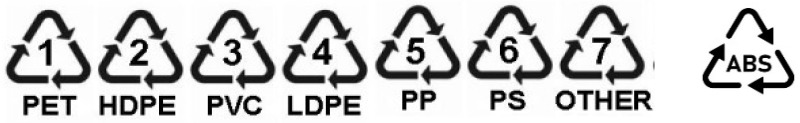
SPI codes.

**Figure 2 polymers-16-01343-f002:**
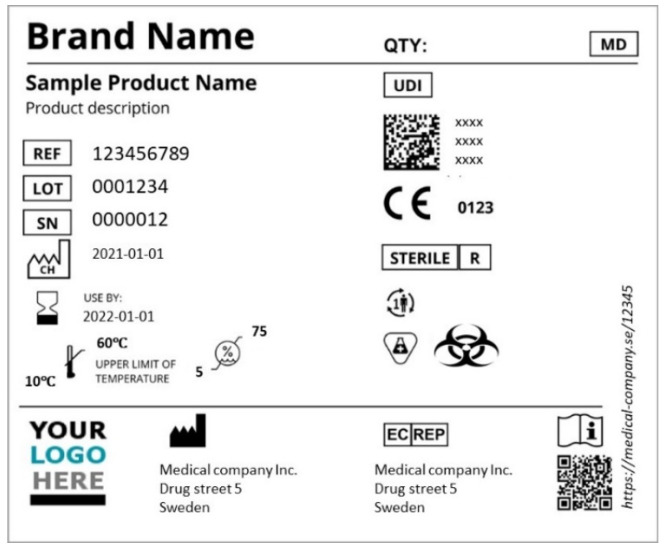
Labelling in accordance with MDR.

**Figure 3 polymers-16-01343-f003:**
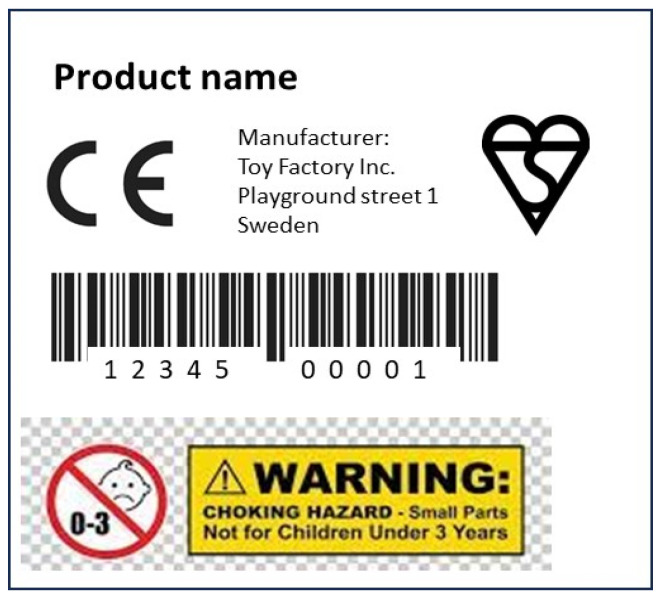
Labelling in accordance with TSD.

**Figure 4 polymers-16-01343-f004:**
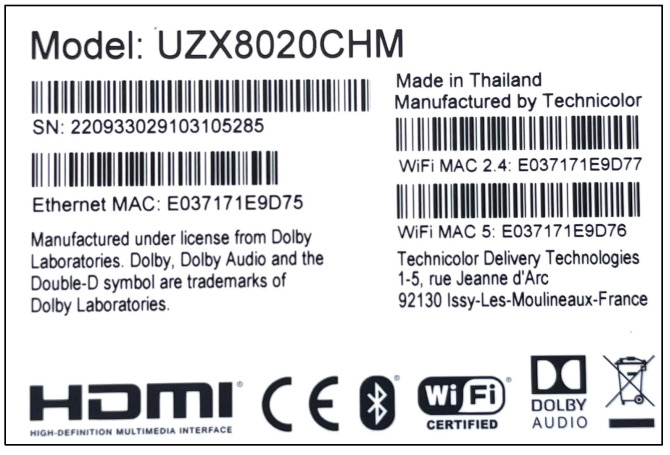
Product traceability labelling for electronic products.

**Table 1 polymers-16-01343-t001:** Summary of the SWOT analysis (based on [30]).

	Strengths	Weaknesses	Opportunities	Challenges
**Physical marker and tracer**	Solution to improve sorting efficiency immediately	Tracer materials remain in plastic	Few technical hurdles for implementation	Single initiatives without a common standard
**Blockchain**	High transparency and security	Energy-intense verification system Risk of false information	The ability to build reliable trust can create incentive opportunities	Many technical hurdles
**Digital Product Passport**	Building on established tools and machinery in line with EU regulations	Non-transparent data handling	Offers link for combining physical and digital solutions	Data ownership and security are unclear
**Certification system**	Can build on established certification systems	Slow implementation with global alignment. No traceability on object level	Can build on global established knowledge	It is unclear as for how to solve the traceability issue and does not cover all aspects or participants of VC

## Data Availability

Data are contained within the article.

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
