# Peer review of "Review and Assessment of Existing and Future Techniques for Traceability with Particular Focus on Applicability to ABS Plastics"

_polymers, 2024, doi:10.3390/polym16101343_

Round 1

Reviewer 1 Report

Comments and Suggestions for Authors

The authors evaluated several existing, emerging, and future traceability and identification techniques, with a primary focus on ABS. The manuscript needs a major revision before eventual publication:

> Abstract. Please define each abbreviation upon first presentation. Make clear the main finding of this review, which procedure is most accurate to identify ABS;

>The introduction should make clear the novelty of the manuscript, the gap in the literature and the reason for focusing on ABS. In addition, a review article needs to explore several references to support the work. For example, the authors used 24 references for a review article;

>Page 2.” The ABS resin is basically a terpolymer consisting of three monomers: acrylonitrile, butadiene, and styrene…..”. Report references addressing the characteristics of ABS;

> Authors could add a topic on methodology to inform the review criteria. The literature review can be classified according to its purpose, scope, function, and analysis developed type. The authors could make clear in the methodology each one of the previous points. What purpose? What is the scope? What is the function? What is the intent?

> Authors must add an exclusive topic on the characteristics of ABS, covering synthesis, properties, applications and resistance to degradation;

>Page 3. Figure 1. Correct by 1 = PET and 3 = PVC;

> Please add a topic demonstrating advantages and disadvantages about ABS identification procedures, cost and feasibility. It would be advisable to review the results of the literature;

>Add a new topic addressing future perspectives for ABS recycling, what can be improved and possible path;

Comments on the Quality of English Language

Moderate editing of English language required

Author Response

Reviewer 1.

Dear Professor,

Thank you for considering our paper for publication. We have now prepared a revised version along the lines suggested by the reviewer. We found several comments useful which helped us to improve our paper. We have also had our article language reviewed by a professional company. We send you the final version as requested with following comments to the referees’ viewpoints (new text in the paper is highlighted in red):

>Page 2.” The ABS resin is basically a terpolymer consisting of three monomers: acrylonitrile, butadiene, and styrene…..”. Report references addressing the characteristics of ABS;

We have included a new chapter (1.1) in the introduction titled “Characteristics of ABS” with few peer-revied reference papers.

> Authors could add a topic on methodology to inform the review criteria. The literature review can be classified according to its purpose, scope, function, and analysis developed type. The authors could make clear in the methodology each one of the previous points. What purpose? What is the scope? What is the function? What is the intent?

A new chapter is included in the introduction (1.4) titled “Aim and methods” which hopefully provides satisfactory answers to the questions.

> Authors must add an exclusive topic on the characteristics of ABS, covering synthesis, properties, applications and resistance to degradation;

The most important information can now be found in chapter 1.1 

>Page 3. Figure 1. Correct by 1 = PET and 3 = PVC;

Figure 1 is corrected and supplemented.

> Please add a topic demonstrating advantages and disadvantages about ABS identification procedures, cost and feasibility. It would be advisable to review the results of the literature;

Full identification of ABS material is costly and does not provide enough information for circular use of ABS. Our study is therefore based on the idea of traceability, which is expected to be able to eliminate the need for identification procedures as complete information about the material will be attached to the product. To review scientific literature about ABS identification procedures, cost and feasibility would require an article of its own.

>Add a new topic addressing future perspectives for ABS recycling, what can be improved and possible path;

A new chapter is included in the introduction (1.2) titled “Recycling of ABS. However, our goal is circular use of ABS made possible by traceability where recycling is only one of several options.

Reviewer 2 Report

Comments and Suggestions for Authors

The authors have identified the important techiques for traceability with focus on ABS. Eventhough it is being positioned as review article by the authors, they need to seriously consider these comments before resubmitting their manuscript for review.

1. For such a review article, the authors neither made a literature review on the research article that talks about the different traceability techniques (need not be specific to ABS)

2. The authors didnt even add the novelty of their literature review

3. They explained many techniques but didnt evaluate which of those traceability techniques could actually be feasible in the present climate

4. Each technique should be accompanied by how the state of the art literature dealth with them

5. Also for a normal reader, the description of different techniques feels like a redundant text and abstract at times. There are not figures on the working of these (some of these) techniques.

6. It is better for authors to evaluate the literature related to the recycling of ABS, their challenges and the recent advancements/legislation focussing only on the high-engineered/hard to recycle plastics

7. The discussion of different techniques, their costs, availability, production of recyclates and market mechanisms have to be thoroughly discussed

8. Neither any search-criteria for the literature review nor any comprehensive literature review on the traceability of plastics has been conducted in this manuscript

Comments on the Quality of English Language

The language reads well. However, a thorough grammar check is once again advisable as there are some long sentences in the text which can be broken down into small sentences.

Author Response

Reviewer 2.

Dear Professor,

Thank you for considering our paper for publication. We have now prepared a revised version along the lines suggested by the reviewer. We found several comments useful which helped us to improve our paper. We have also had our article language reviewed by a professional company. We send you the final version as requested with following comments to the referees’ viewpoints (the new text in the paper is highlighted in red):

  1. For such a review article, the authors neither made a literature review on the research article that talks about the different traceability techniques (need not be specific to ABS)

There are several publications about new data communication systems for traceability, but they are mostly of a general nature or deal with other industries such as food supply. The capability of the Digital Product Passports and Blockchain technology within the plastics sector are emerging and future research and adaptation is needed to explore their use by various plastic value chains. This paper is the first step in our project with the aim to review and evaluate existing and emerging traceability technologies that could be suitable to use in ABS products value chains. Based on the information in this paper, we will prepare at a later stage a guideline on how a traceability system for ABS products should be designed.

  1. The authors didn’t even add the novelty of their literature review

The novelty of this study is highlighted in the new chapter in the introduction (1.2) titled “Aim and methods”.

  1. They explained many techniques but didn’t evaluate which of those traceability techniques could actually be feasible in the present climate

This is now explained in Abstract and in chapter 5. “Concluding remarks”.

  1. Each technique should be accompanied by how the state of the art literature deal with them

This investigation has shown that the scientific literature is of general and principled character, but we have not found any research reports with information related to ABS.

  1. Also for a normal reader, the description of different techniques feels like a redundant text and abstract at times. There are not figures on the working of these (some of these) techniques.

In our opinion, for those interested in research, development and design of future traceability systems for plastics and especially ABS products, the information on the available technologies' basic principles and their strengths and weaknesses should be useful as it is useful for our further work in ABSolEU project.

  1. It is better for authors to evaluate the literature related to the recycling of ABS, their challenges and the recent advancements/legislation focussing only on the high-engineered/hard to recycle plastics

A new chapter is included in the introduction (1.2) titled “Recycling of ABS with some examples of research into different recycling methods. However, our goal is circular use of ABS made possible by traceability where recycling is only one of several options. In our opinion traceability is expected to facilitate and improve recycling but above all enable circular use of ABS.

  1. The discussion of different techniques, their costs, availability, production of recyclates and market mechanisms have to be thoroughly discussed

As there are no traceability systems in use for ABS products, there is also no information on costs, availability and market mechanisms available. Which techniques are suitable, their cost and market mechanisms are all important questions that will be investigated for different ABS products in the next part of our research.

  1. Neither any search-criteria for the literature review nor any comprehensive literature review on the traceability of plastics has been conducted in this manuscript

A new chapter (1.4) which contains information about search methods and aim is included in the introduction titled “Aim and methods”. There is a rather limited scientific literature dealing with traceability specifically adapted to plastic products and we couldn´t find any paper dealing with traceability linked to ABS plastics.

Round 2

Reviewer 1 Report

Comments and Suggestions for Authors

The authors have improved the quality of the manuscript in this revised version. The recommendations were accommodated in the manuscript, thus improving the degree of clarity.

Comments on the Quality of English Language

Minor editing of English language required

Author Response

Dear Reviewer,

 Thank you for reviewing our manuscript.

Reviewer 2 Report

Comments and Suggestions for Authors

The authors have taken efforts in addressing some of the comments raised by the reviewers. Still, I feel there are some major revisions that have to be done before the manuscript can be considered for the acceptance. The comments are as follows:

1. I asked for a literature review for the traceability of plastics in general. That is still missing in Section 2. If the authors say their manuscript reviews existing and future techniques (although with a focus on ABS), I woud like to know what the literature says when it comes to the traceability. It can be for other plastics, sectors or for othe rproducts. If you submit the revised version without a literature review of traceability of plastics in research, unfortunately I have to reject the manuscript

2. The second paragraph of 1.4 shows the authors have not followed a specific set of methodlogical approaches when it comes to the literature search. Saying that the assessment was carried out using books, papers or Scopus is not enough. For a review paper, it is important to show the time frame you set for your search, how many results you have got, how many literature sources you have considered for review (need not be only for ABS)., what were those sources, what kind of techniques were searched in the literature review, how you finalised on these traceability techniques and so on. These things are missing in the manuscript

3. You have mentioned that there are not diagrams exist for traceability techniques for ABS, but if you are saying a technology is ideal for the traceability, the least I would like to know is to understand how that technology works when it comes to recycling of ABS. I once again emphasise on the visual representation of the important traceability techniques (if not all)

4. It is better to cite a scientific publication when talking about Blockchain technology (when you position it as an ideal technology for ABS) and talk about the challenges and beneifts of it over other technologies. Also cite in the conclusion where EU commission have pointed out Blockchain as the future.

Author Response

 Dear Reviewer,

Please see our answer in attached file.

Round 3

Reviewer 2 Report

Comments and Suggestions for Authors

Appreciate the authors' efforts to address the comments. The manuscript can now be accepted.